# Compound 21, a two-edged sword with both DREADD-selective and off-target outcomes in rats

**Raphaël Goutaudier, Véronique Coizet, Carole Carcenac, Sebastien Carnicella** [ORCID] *

Institut national de la santé et de la recherche médicale, Grenoble Institut des Neurosciences, U1216, Université, Grenoble Alpes, Grenoble, France

* sebastien.carnicella@inserm.fr

## Abstract

Designer Receptors Exclusively Activated by Designer Drugs (DREADDs) represent a technical revolution in integrative neuroscience. However, the first used ligands exhibited dose-dependent selectivity for their molecular target, leading to potential unspecific effects. Compound 21 (C21) was recently proposed as an alternative, but *in vivo* characterization of its properties is not sufficient yet. Here, we evaluated its potency to selectively modulate the activity of nigral dopaminergic (DA) neurons through the canonical DREADD receptor hM4Di using *TH-Cre* rats. In males, 1 mg.kg$^{-1}$ of C21 strongly increased nigral neurons activity in control animals, indicative of a significant off-target effect. Reducing the dose to 0.5 mg.kg$^{-1}$ circumvented this unspecific effect, while activated the inhibitory DREADDs and selectively reduced nigral neurons firing. In females, 0.5 mg.kg$^{-1}$ of C21 induced a transient and residual off-target effect that may mitigated the inhibitory DREADDs-mediated effect. This study raises up the necessity to test selectivity and efficacy of chosen ligands for each new experimental condition.

## Introduction

Designer Receptors Exclusively Activated by Designer Drugs (DREADDs) are chemogenetic tools that represent one of the major breakthroughs of the last ten years in integrative neuroscience [1,2]. Combining the precision of genetics with pharmacology, DREADDs provide a remote, prolonged and reversible control of neuronal or extra-neuronal subpopulations via conditional expression and allow the study of complex phenomena in awake animals. As such, they were elegantly used to easily induce tonic modulation, affording an alternative to optogenetics which is more adapted for phasic modulation [3], and to study the implication of different neural system in various behaviors such as feeding, memory, pain or motivation (reviewed in [4]). Initially described as a "lock and key" system, DREADDs are G-protein-coupled receptors that rely on the combination of a mutated muscarinic receptors, that have lost their affinity for acetylcholine, and a designed drug which binds to the mutated receptor with potentially otherwise no pharmacological activity [5]. Two Designed Receptors were originally and are

**Funding:** This work was supported by the Institut National de la Santé et de la Recherche Médicale

(Inserm), the Agence Nationale de la Recherche (ANR-16-CE16-0002, to SC) and Grenoble Alpes University.

**Competing interests:** The authors have declared that no competing interests exist.

commonly used for DREADD modulation: hM3Dq, coupled to Gq protein which increases neuronal activity and hM4Di, coupled to Gi protein which decreases neuronal activity. A next generation of DREADDs deriving from other endogenous metabotropic and ionotropic receptors were developed over time [6]. Similarly, different DREADDs ligands have been developed. The first DREADD ligand was Clozapine-N-oxide (CNO), a derived metabolite of the atypical antipsychotic clozapine. Initially described to be devoid of endogenous activity at moderate doses [5], this ligand was widely used to activated DREADDs during the past decade. However, the selectivity of this compound depend on the dose (e.g., [7]), and it was observed that CNO induced behavioral off-target effects in both mice and rats which did not express DREADDs [8,9]. In addition, Gomez et al. [10], reported that CNO was not the real DREADDs activator since it was not able to cross the blood brain barrier (BBB) and was in fact back-metabolized in low doses of clozapine. However, behavioral investigations quickly showed that even low doses of clozapine induce anxiety-related behaviors in naïve animals [11,12], indicating that this molecule is not appropriated as a DREADDs ligand (see also [3]). All these observations together led to the necessity of developing new ligands. As such, a second generation of ligands was engineered leading to the creation of three new synthetic ligands: compound 21 (C21) developed by Chen et al. [13], JHU37152 and JHU37160 (JHUs) developed by Bonaventura et al. [14]. Compared to JHUs, C21 has been developed earlier and as such, has gained interest in the field. For instance, Thompson et al. [15], have demonstrated that C21 from 0.3 to 3 mg. $kg^{-1}$ was sufficient to activate DREADDs in mice and to induce selective behavioral alterations. C21 appears therefore to be an interesting DREADDs activator. However, it remains to be fully characterized in other species and other experimental conditions, as caution is needed since designed ligands could have different outcomes depending on the doses, species, strains or gender used [3]. In the present study, we aimed at further document the *in vivo* properties of C21 by extending its DREADDs application to transgenic *TH-Cre* rats. Indeed, *TH-Cre* rats are frequently used in combination to DREADDs as they appear as a powerful tool for the investigation of tonic modulation of mesolimbic and nigrostriatal dopaminergic (DA) systems in motivated, cognitive and affective behaviors [16–19]. However, no one has tested yet the potential efficiency and selectivity of C21 in this experimental model.

## Materials and methods

### Animals

29 males and 13 females *TH-Cre* rats (breeding at the Plateforme Haute Technologie Animal, La Tronche) were included in this study. They were housed in a 12h/12h reverse light cycle, with food and water *ad libitum.* At the beginning of the experiments, the males weighed between 240 and 430g and females weighed between 200g and 320g. All experimental protocols complied with the European Union 2010 Animal Welfare Act and the new French directive 2010/63, and were approved by the French national ethics committee no. 004.

### Stereotaxic viral infusion

Animals were anesthetized with a mixed intraperitoneal (i.p.) injection of ketamine (Chlorkétam, 60 mg.$kg^{-1}$, Mérial SAS, Lyon, France) and xylazine (Rompun, 10 mg.$kg^{-1}$, Bayer Santé, Puteaux, France). Then local anesthesia was provided by a subcutaneous injection of lidocaïne (Lurocaïne, 8 mg. $kg^{-1}$, Laboratoire Vetoquinol S.A., France) on the skull surface and animal were secured in a Kopf stereotaxic frame under a microbiological safety post (PSM). Coordinates for SNc injections were determined according to [20], adjusted to the body weight and set at, relative to bregma: -4.3 mm (AP), ±2.4 mm (ML), -7.9 mm (DV). Animals were infused bilaterally with 1 µl of AAV5-hSyn-DIO-hM4Di-mCherry ($10^{12}$ particles.$ml^{-1}$, Addgene,

Watertown, Massachusetts, États-Unis, #44362-AAV5) or 1 μl of AAV5-hSyn-DIO-mCherry ($10^{12}$ particles.ml$^{-1}$, Addgene, #50459-AAV5). The virus was infused at a rate of 0.2 μL.min$^{-1}$ using microinjection cannula (33-gauge, Plastic One, USA) connected to a 10 μL Hamilton syringe and a microinjection pump (Stoelting Co., Wood Dale, IL). After injection, the cannula remained *in situ* for 5 min before withdrawal to allow the injected solution to be absorbed into the parenchyma. The skin was sutured, disinfected, and the animal placed in a heated wake-up cage, before being replaced in its home-cage after complete awakening and monitored for a couple of days.

## Reagent

C21 (Hello Bio, Bristol, UK) was dissolved in 0.9% saline and kept at -20°C before testing. All the injections were given intraperitoneally, at 0.5 or 1 mg.kg$^{-1}$ (at a volume of 1 mL.kg$^{-1}$). A vehicle solution (NaCl 0.9%) was prepared and kept in the same conditions.

## *In vivo* extracellular electrophysiology

At least two weeks after viral infusion, we performed extracellular multiunit recordings to assess neuronal activity of SNc neurons. Rats were anesthetized continuously with isoflurane and body temperature was maintained at 37°C with a thermostatically controlled heating blanket. Two tungsten electrodes (Phymep, Paris, France), allowing recording of a neuronal population, were implanted bilaterally into the SNc using the coordinates determined according to [20], and set at: -4.3 mm (AP, bregma), ±2.3 mm (ML, bregma) and -6.5 mm (DV, brain surface). Coordinates between infusion and electrophysiology were slightly changed to avoid the area of mechanical injury induced by the injection. Extracellular voltage excursions were amplified, band-pass filtered (300 Hz–10 kHz), digitized at 10 kHz and recorded directly onto computer disc using a Micro 1401 data acquisition system (Cambridge Electronic Design [CED] Systems, Cambridge, UK) running CED data capture software (Spike 2). Once electrodes were implanted and signal stabilized, baseline (BL) without treatment was recorded during 10 minutes before i.p. administration of a vehicle solution (VEH—NaCl 0.9%). The VEH period of 20 minutes was followed by i.p. administration of C21 (1 mg.kg$^{-1}$ or 0.5 mg.kg$^{-1}$), for a recording period of 240 minutes. Then, the position of SNc recording sites were marked with a small lesion caused by passing 10 μA DC current for 1 min through the tungsten recording electrode. Multi-unit activity was normalized by the baseline activity of the first 10 minutes. Recordings with more than 25% of variation of the multi-unit activity between the baseline pre-injection and the vehicle periods were excluded from the study. One recording per hemisphere were performed. Recordings were excluded after histological and immunohistological analyses (see below) when the recording site was outside the SNc and/or when DREADDs or control virus expression in the SNc was absent.

As SNc contains of a majority of DA neurons but also a minority of GABA neurons [21], to decipher the nature of the neurons recorded we performed additional spike analysis, focusing on the action potential's shape. In multi-unit recordings, DA neuronal extracellular signals are characterized by a triphasic spike, with a duration greater than 2 ms, and a duration measured from spike initiation to the maximal negative phase of the action potential greater than 1.1 ms [22–24]. Only recording fulfilling these criteria where included in the study. In addition, we observed a low variability in the shape of the waveform average obtained from our recordings indicating that this average is from a highly homogeneous population of putative dopaminergic neurons with a long duration triphasic action potentials, and is unlikely to include GABA neurons with biphasic short-lasting spikes (Fig 1).

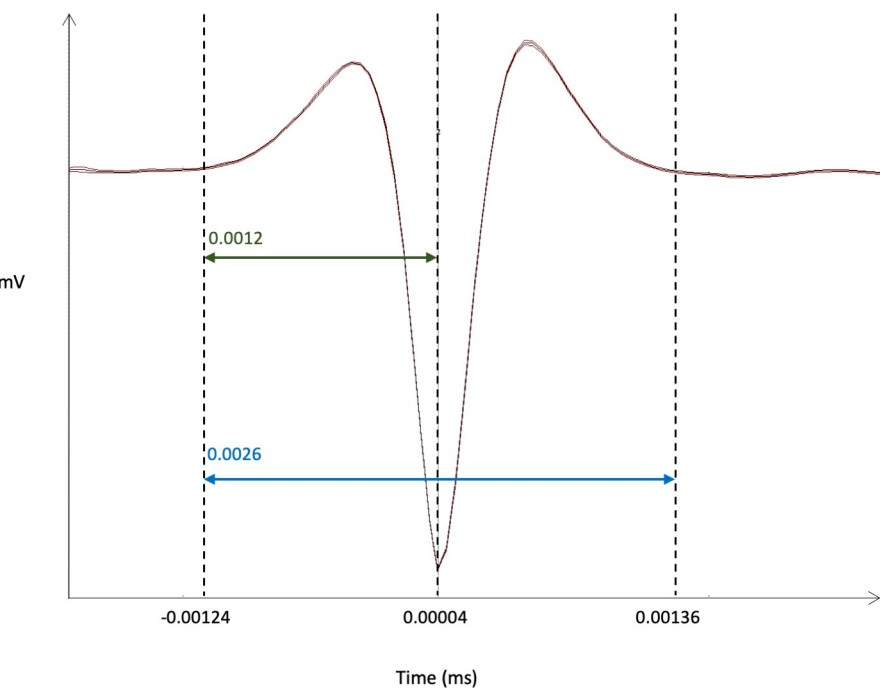

**Fig 1. Representative example of the waveform average of recordings.** The total duration of the triphasic spike is indicated in blue. The duration, from spike initiation to the maximal negative phase of action potential, is indicated in green. Data are presented as the waveform average +/- SEM.

## Tissue preparation and histological validation

At the end of the experiment, rats were deeply anesthetized by isoflurane saturation and transcardially perfused with 0.9% NaCl (100 mL) followed by 4% paraformaldehyde (300 mL, PFA) in phosphate-buffered saline (PBS). After decapitation, brains were extracted and post- fixed for 24h in 4% PFA. They were then cryoprotected in 20% sucrose/PB for 24h and frozen in isopentane cooled to -50˚C on dry ice. Coronal sections (30 μm) of mesencephalon were cut using a cryostat (Microm HM 525; Microm, Francheville, France). Placement of electrodes were verified by Cresyl violet staining and visualized with the ICS FrameWork computerized image analysis system (TRIBVN, 2.9.2 version, Châtillon, France), coupled to a light microscope (Nikon, Eclipse 80i) and a Pike F-421C camera (ALLIED Vision Technologies, Stadtroda, Germany) for digitalization. Meanwhile, floating coronal section of three levels of the mesencephalon were selected as previously described [25] for assessment of DREADDs expression.

## TH-immunohistochemistry and DREADDs expression localization

To assess DREADDs expression in DA mesencephalic regions, immunostaining for tyrosine hydroxylase (TH) was performed. Free-floating 30 μm thick coronal sections were washed with TBS and incubated for 1 h in 0.3% Triton X-100 in TBS (TBST) and 3% normal goat serum (NGS). They were then incubated with primary monoclonal mouse anti-TH antibody (mouse monoclonal MAB5280, Millipore, France, 1/2500) diluted in TBST containing 1% NGS overnight (4˚C). Then, slices were incubated with a green fluorescent conjugated goat anti-mouse Alexa 488 antibody (1/500, Invitrogen™, Waltham, Massachusetts, USA) for 1h30 at room temperature. They were finally mounted on superfrost glass slides, with Aqua-Poly/Mount (Polysciences, Inc., Germany). Fluorescent pictures of TH labelling and mCherry

expression were taken using a slide scanner (Z1 Axioscan, Zeiss Göttingen, Germany), at x20 magnification and analyzed with ImageJ. DREADD expression was quantified for each hemisphere by comparing the number of TH-labeled-mCherry-positive neurons with the number of TH-labeled neurons within three areas: the lateral SNc (lSNc), the medial SNc (mSNc) and the Ventral Tegmental Area (VTA). We have also verified first that mCherry was only detected in TH-positive neurons. Fluorescent illustrations presented in this article were taken with a laser-scanning confocal microscope (LSM710, Zeiss,). Z-stacks of digital images were captured using ZEN software (Zeiss).

### Data analyses

For DREADDs expression, data were expressed as the mean number of quantified hemispheres for which recordings were included +/- SEM (number of quantified hemispheres and animals for each group are detailed in figure legends). For extracellular electrophysiology, data were expressed as the mean number of recordings +/- SEM (number of included recordings and animals for each group are detailed in figure legends). Parametric analyses were performed after verification of the assumptions of normality (Shapiro-Wilk and Kolmogorov-Smirnov tests) and sphericity (Bartlett's test). Data were analyzed by t-test, RM one-way ANOVA, two-way ANOVAs and RM two-way ANOVAs, depending on the experimental design, using GraphPad Prism 8 (summarized in S1 Table). As the electrophysiological recordings were long, some values were missing due to artefacts (3% of the data recorded in extracellular electrophysiology). In this case, data were analyzed by fitting a mixed model proposed by the statistical software. This mixed model uses a compound symmetry covariance matrix, and is fit using Restricted Maximum Likelihood (REML). When indicated, post hoc analyses were carried out with the Bonferroni's correction procedure. Significance for p values was set at $\alpha =$ 0.05. Effect sizes for the ANOVAs were also reported using partial $\eta^2$ values [26,27]. Determining these values from the mixed-model analysis was however not accessible.

### Results

We thereby infused into the substantia nigra pars compacta (SNc) of male and female *TH-Cre* rats, a floxed virus encoding for the inhibitory DREADDs hM4Di coupled to mCherry (♂-*hM4Di* and ♀-*hM4Di*). Meanwhile, a floxed virus encoding only for mCherry was infused in control groups (♂-*mCherry* and ♀-*mCherry*) (Fig 2). With extracellular electrophysiology, we first verified that basal activity of the neuronal subpopulations recorded within the SNc were comparable between our different experimental groups, since DREADDs may have a constitutive activity [28]. We found neither differences between ♂-*hM4Di* and ♂-*mCherry* animals (23.6 events/s ± 1.9 and 21.2 events/s ± 2.9 respectively, t(39) = 0.65, p = 0.521) nor between ♀-*hM4Di* and ♀-*mCherry* animals (38.8 events/s ± 7.3 and 26.7 events/s ± 4.3 respectively, t(13) = 1.48, p = 0.164). We also verified that basal activity of this multi-unit recording remains stable over time in groups of mCherry and hM4Di animals only treated with saline (RM one-way ANOVAs report no effect of time: $F_s < 0.98$, p > 0.46, partial $\eta^2 < 0.12$; S1 Fig).

Then, we tested the potential effect of two concentrations of C21, 1 mg.kg$^{-1}$ and 0.5 mg.kg$^{-1}$, on the neuronal activity of the SNc in ♂-*mCherry* or ♂-*hM4Di* animals (Figs 3 and S2). We first assessed the effect of 1 mg.kg$^{-1}$ of C21 in ♂-*mCherry* animals (Fig 3B and 3D). We observed a robust and persistent increase in the activity of nigral neurons, indicating that, even within the recommended range of doses [15], C21 can have strong non-DREADDs mediated pharmacological effects. Importantly, decreasing the dose of C21 to 0.5 mg.kg$^{-1}$ allowed to completely circumvent this unspecific effect on SNc neuronal activity (Mixed-effects analysis highlights a main effect of treatment: $F_{(1, 13)} = 16.08$, p < 0.01; of the time: $F_{(8, 98)} = 6.38$,

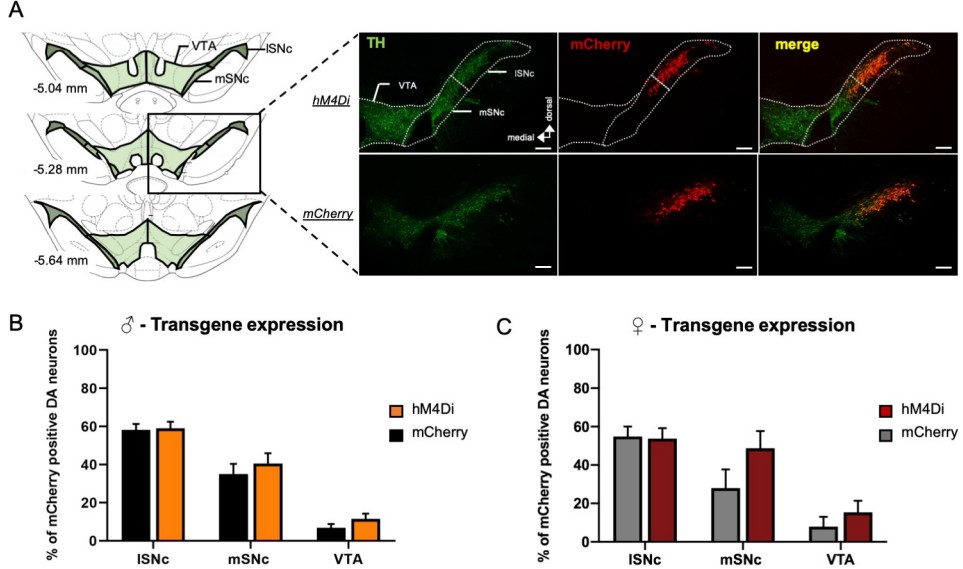

**Fig 2. hM4Di-mCherry and mCherry expression in mesencephalic DA neurons of *TH-Cre* rats. (A)** On the left, schema of the three levels of mesencephalon used for quantified viral expression with three areas: lateral SNc (lSNc), medial SNc (mSNc) and Ventral Tegmental Area (VTA). The black rectangle indicates the level at which representative images, on the right, were taken to illustrate TH immunostaining and hM4Di-mCherry or mCherry expression. **(B-C)** Percent of transgenes expression in the lSNc, mSNC and VTA for males **(B)** (hM4Di, orange, n = 18 hemispheres, 13 animals; mCherry, black, n = 23 hemispheres, 16 animals) or females **(C)** (hM4Di, red, n = 7 hemispheres, 7 animals; mCherry, grey, n = 8 hemispheres, 6 animals). A similar pattern of expression was observed between males and females with a gradient of expression from lSNc to the VTA (two-way ANOVAs: Fs > 18.83, $p < 0.001$, partial η2 > 0.49), in both transgene conditions (two-way ANOVAs: Fs < 2.45, $p > 0.16$, partial η2 < 0.06). For each area, the given percent of expression correspond to the mean expression of the three levels of mesencephalon. Scale bar: 200 μm. Data were expressed as the mean number of quantified hemispheres for which recordings were included +/- SEM.

$p < 0.001$; and treatment × time interaction: $F_{(8, 98)} = 5.44$, $p < 0.01$; Fig 3D, *left panel*). Over-all, a 100%-increase was observed between 90 to 180 minutes after the injection of C21 at 1 mg.kg$^{-1}$ compared to vehicle, an effect that was absent at 0.5 mg.kg$^{-1}$ (Two-way ANOVAs showed a main effect of the treatment: $F_{(1, 13)} = 17.45$, $p < 0.01$, partial $\eta^2 = 0.57$; of the transgene: $F_{(1, 13)} = 17.81$, $p < 0.01$, partial $\eta^2 = 0.59$; and treatment x transgene interaction: $F_{(1, 13)} = 15,74$, $p < 0.01$, partial $\eta^2 = 0.55$; Fig 3D, *middle and right* panel). We next tested whether the dose of 0.5 mg.kg$^{-1}$ of C21, devoid of off-target effect in the SNc, was sufficient to activate the DREADDs in ♂-*hM4Di* animals and to produce significant *in vivo* chemogenetic effects. As shown on Fig 3C and 3E, 0.5 mg.kg$^{-1}$ of C21 induced in ♂-*hM4Di* but not in ♂-*mCherry* animals, a significant reduction of SNc neuronal activity, as expected from the activation of an inhibitory receptor selectively expressed in a DA neuronal subpopulation. This decrease became evident 90 minutes after the injection of C21 and ended 120 minutes later (Mixed-effects analysis highlighted a main effect of transgene: $F_{(1,17)} = 5.89$, $p < 0.05$; marginal effect of time $F_{(8,130)} = 1.97$, $p = 0.055$ and no significant transgene x time interaction $F_{(8,130)} = 1.67$, $p = 0.113$; Fig 3E, *left panel*). Overall, a 30%-decrease was observed between 90 to 180 minutes after injection of 0.5 mg.kg$^{-1}$ of C21 compared to vehicle and the mCherry control condition (Two-way ANOVAs showed a main effect of the treatment: $F_{(1, 17)} = 6.52$, $p < 0.05$, partial $\eta^2 = 0.28$; of the transgene: $F_{(1, 17)} = 6.69$, $p < 0.05$, partial $\eta^2 = 0.38$; and treatment x transgene interaction: $F_{(1, 17)} = 10.26$, $p < 0.01$, partial $\eta^2 = 0.37$; Fig 3E, *middle and right panel*). Notably, we also observed a complete recovery of the basal activity in ♂-*hM4Di* rats at 240 minutes post-injection. This indicates that, consistently with the DREADDs approach, this effect was

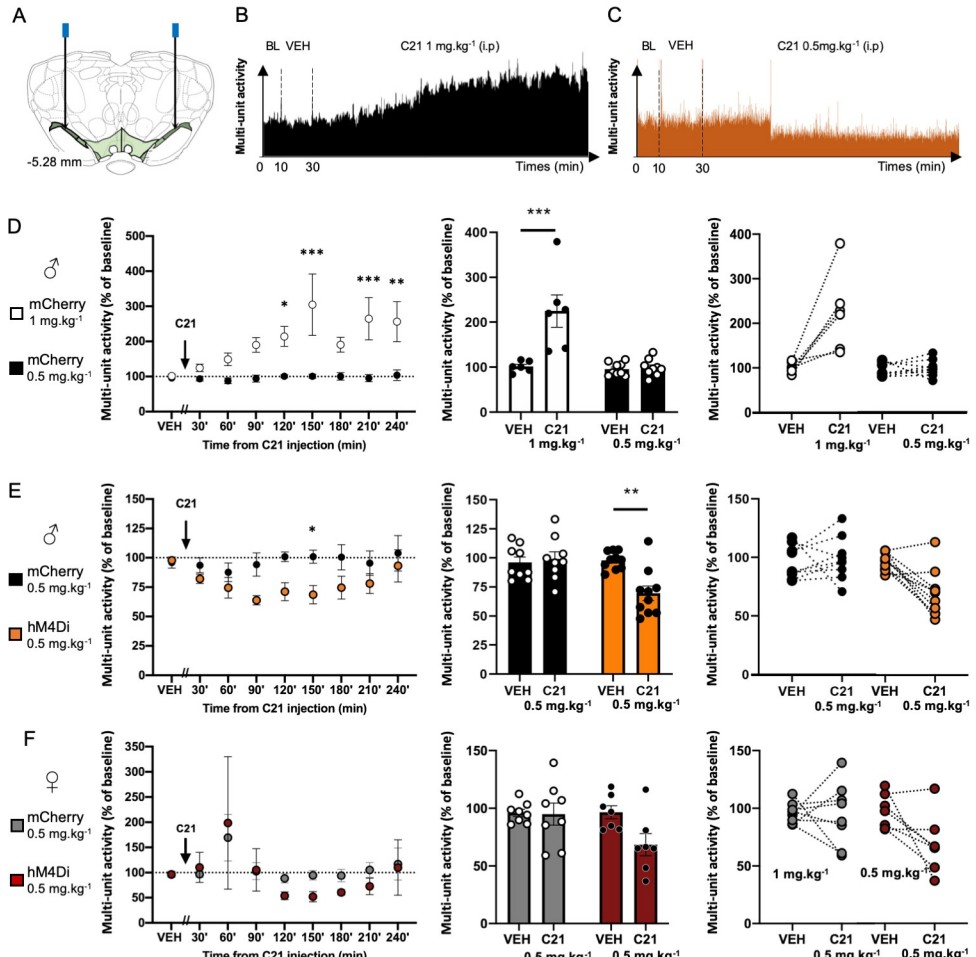

**Fig 3. Dose dependent effect of C21 on multi-unit activity of SNc neurons expressing hM4Di-mCherry or mCherry. (A)** Schema of the bilateral electrodes implantations. **(B)** Representative data obtained during recording of neuronal subpopulation within the SNc from male mCherry rat treated with 1 mg.kg$^{-1}$ of C21. **(C)** Representative data obtained during recording of neuronal subpopulation within the SNc from male hM4Di rat treated with 0.5 mg.kg$^{-1}$ of C21. **(D-F)** *On the left*, effect of C21 along time on SNc neuronal multi-unit activity, during vehicle (VEH, a 20-minutes interval) and C21 periods (30-minutes intervals), normalized to 10-minutes baseline recording. *In the middle and on the right*, mean neuronal multi-unit activity, during the VEH period and between the 90 and 180 minutes post-C21 injection intervals, normalized to baseline. **(D)** Effect of C21 in mCherry male rats treated with 1 (white, n = 6 recordings, 4 animals) or 0.5 (black, n = 9 recordings, 7 animals) mg.kg$^{-1}$. **(E)** Effect of C21 in mCherry (black, n = 8 recordings, 7 animals) or hM4Di (orange, n = 10 recordings, 8 animals) male rats treated with 0.5 mg. kg$^{-1}$. **(F)** Effect of C21 in mCherry (grey, n = 8 recordings, 6 animals) or hM4Di (red, n = 7 recordings, 7 animals) female rats treated with 0.5 mg.kg$^{-1}$. Data were expressed as the mean number of recording sides +/- SEM. BL: baseline, VEH: vehicle. *P < 0.05, **P < 0.01, ***P < 0.001.

reversible, and not due to a loss of signals along time. These finding indicate that, in male *TH-Cre* rats, 0.5 mg.kg$^{-1}$ of C21 is sufficient to potently activate hM4Di in *TH-Cre* rats, without inducing endogenous off-target effects.

As brain responses may differ between males and females [29], we next investigated the effect of 0.5 mg.kg$^{-1}$ of C21 in female *TH-Cre* rats. As in male, this dose of C21 selectively decreased the activity of nigral neurons in ♀-*hM4Di* rats (Fig 3F). This effect was however not detected as significant (Fig 3F, *left panel*: Mixed-effects analysis report no significant effect of the transgene: $F_{(1, 13)}$ = 0.28, p = 0.605; neither of time: $F_{(8, 95)}$ = 1.88, p = 0.072; nor time x transgene interaction: $F_{(8, 95)}$ = 0.23, p = 0.977; Fig 3F, *middle and right panel*: Two-way

ANOVAs report a marginal effect of treatment: $F_{(1,13)}$ = 3.63, p = 0.079, no significant effect of the transgene: $F_{(1, 13)}$ = 3.13, p = 0.1, or significant treatment x transgene interaction: $F_{(1, 13)}$ = 3.13, p = 0.1), consistent with lower effect sizes of DREADDs as compared to male (Fig 3F, *middle and right panel*: partial $\eta^2$ = 0.22, partial $\eta^2$ = 0.18, partial $\eta^2$ = 0.19, for the treatment, the transgene an treatment x transgene interaction respectively). This is probably due to the fact that, in contrast to male, a residual unspecific effect of C21, highlighted by a clear transient increase of 169% of the activity at 60 min post-injection (Fig 2F), likely mitigated the following DREADDs-mediated effect. (Fig 3F, *left panel*). Therefore, detecting a statistically significant DREADDs-mediated effect may require a greater number of animals for female with this dose of C21.

## Discussion

Here, we demonstrated that C21 possesses both specific and unspecific effect on rats depending on doses used. In males, at 0.5 mg.kg$^{-1}$, C21 activated hM4Di with a potent *in vivo* effect, without inducing off-target effect. This led to a reversible inhibition of nigral neurons activity selectively in hM4Di-expressing animals. Conversely, at 1 mg.kg$^{-1}$, C21 induced a robust and long-lasting increase of SNc neurons activity in hM4Di-lacking animals. In females, this unspecific effect was also transiently observed, in both hM4Di-expressing and hM4Di-lacking animal, with the dose of 0.5 mg.kg$^{-1}$, meaning that precaution must be taken in studies using both genders. This is critical because scientists working on transgenic lines often used males and females to obtain larger cohorts (e.g., [10,14,28,30]). Relative potent affinity of C21 for some endogenous receptors may account for the off- target effect evidenced in the present study. Indeed, C21 may exhibit similar affinity for serotoninergic 5-HT2 Gi-coupled and histaminergic H1 Gq-coupled receptors than for hM4Di behaving potentially as a competitive antagonist of these receptors [3,7,15]. Given that 5-HT2 Gi-coupled receptors are expressed on SNc DA neurons [31,32], by blocking this inhibitory receptor, C21 can promote SNc neurons activity [33]. In addition, blocking H1 Gq-coupled receptors that are located on nigral GABAergic neurons can lead to a reduction of the GABAergic inhibition on nigral DA neurons and therefore enhance their activity [34,35]. Although these two hypotheses remain speculative and deserve further investigations, it appears not unlikely that, depending on the dose, C21 exhibits such off-target effect on SNc neuronal activity.

Finally, this study demonstrates that C21 can be a potent DREADDs activator in rats. It also clearly illustrates that, because DREADDs derive from endogenous receptors and rely on the use of pharmacological compounds, they are unlikely to be fully devoid of off-target effects, even if new ligands are proposed each year and help to maximize this approach. These effects will always depend on the dose, the species, the strains and the gender used. Therefore, regardless of the chosen ligand, a "model-dependent" approach must be adopted to assess the selectivity and efficiency of the ligand for every new experimental condition prior any behavioral experiment.

## Supporting information

**S1 Fig. Basal neuronal activity and recording remain stable over time in animals treat with saline solution.** Effect of vehicle along time on SNc neuronal activity rate in rats expressing mCherry (n = 8 recordings, 5 animals) **(A)** or hM4Di (n = 8 recordings, 5 animals) **(B)**. To keep the same experimental conditions, two saline injections were realized, one at the end of the 10-minutes baseline recording (VEH, a 20-minutes interval) and one at the end of the vehicle period corresponding to the time of C21 injection (30-minutes intervals). Data were

expressed as the mean number of recording sides +/- SEM.
(TIF)

**S2 Fig. Location of recording sites within the SNc among the different groups studied.**
Male expressing mCherry treated with 1 mg.kg$^{-1}$ of C21 (**A**) or 0.5 mg.kg$^{-1}$ of C21 (**B**). Male
expressing hM4Di treated with 0.5 mg.kg$^{-1}$ of C21 (**C**). Female expressing mCherry (**D**) or
hM4Di (**E**) and treated with 0.5 mg.kg$^{-1}$ of C21.
(TIF)

**S1 Table. Summary of statistical analyses.**
(TIF)

# Acknowledgments

The authors would like to thank Sabrina Boulet and Yvan Vachez for critical reading of the
manuscript. The authors also would like to thank Jacques Brocard and the PIC GIN Platform
for technical assistance in fluorescence microscopy and analysis, as well as the in vivo experi-
mental platform.

# Author Contributions

**Conceptualization:** Raphaël Goutaudier, Sebastien Carnicella.

**Data curation:** Raphaël Goutaudier.

**Formal analysis:** Raphaël Goutaudier, Véronique Coizet, Sebastien Carnicella.

**Funding acquisition:** Sebastien Carnicella.

**Investigation:** Raphaël Goutaudier.

**Methodology:** Raphaël Goutaudier, Véronique Coizet, Carole Carcenac.

**Resources:** Véronique Coizet.

**Supervision:** Sebastien Carnicella.

**Validation:** Véronique Coizet, Carole Carcenac, Sebastien Carnicella.

**Writing – original draft:** Raphaël Goutaudier, Véronique Coizet, Carole Carcenac, Sebastien
Carnicella.

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
