## [Decision Letter · Decision Letter 0]

24 Jun 2020

PONE-D-20-14375

Compound 21, a two-edged sword with both DREADD-selective and off-target outcomes in rats

PLOS ONE

Dear Dr. Carnicella,

Thank you for submitting your manuscript to PLOS ONE. After careful consideration, we feel that it has merit but does not fully meet PLOS ONE’s publication criteria as it currently stands. In particular, it will be necessary to (1) unequivocally identify the recorded neurons as dopaminergic and (2) respond to the issues concerning statistical power and analyses. Therefore, we invite you to submit a revised version of the manuscript that addresses the points raised during the review process. 

We look forward to receiving your revised manuscript.

Kind regards,

Gilberto Fisone

Academic Editor

PLOS ONE

Journal Requirements:

Additional Editor Comments (if provided):

The reviewers have

Reviewers' comments:

Reviewer's Responses to Questions

**Comments to the Author**

1. Is the manuscript technically sound, and do the data support the conclusions?

Reviewer #1: No

Reviewer #2: Yes

2. Has the statistical analysis been performed appropriately and rigorously? 

Reviewer #1: Yes

Reviewer #2: No

3. Have the authors made all data underlying the findings in their manuscript fully available?

Reviewer #1: Yes

Reviewer #2: Yes

4. Is the manuscript presented in an intelligible fashion and written in standard English?

Reviewer #1: Yes

Reviewer #2: Yes

5. Review Comments to the Author

Reviewer #1: In the past decade, the neurosciences has enjoyed a fascinating development of new tools for remotely controlling neuronal activity in freely moving animals. The DREADD system has received a lot of interest because of its simple application to behavioral assays, making it a potentially wonderful technology accessible to many laboratories in the field. Recently, concerns have emerged about the specificity of the inert ligands used to activate the DREADD receptors, casting some doubt on the validity of the results obtained with the DREADD technology.

In the present study, Goutaudier et al have assessed the actions of a relatively new ligand (C21) in a well-characterized system for in vivo recordings: the midbrain dopamine system; and whether systemic treatment with C21 has the predicted effect on dopamine neurons expressing the inhibitory DREADD receptor hM4Di.

My major concern about this study is whether the authors in fact record from dopamine neurons or not. There is a huge literature on the firing rate of dopamine neurons and the basal firing rate reported is unambiguously between 2-6Hz, depending on the level of anesthesia. In contrast, Goutaudier et al recorded from cells that fire at >20Hz and most likely represent GABA producing interneurons in the midbrain. It is unclear to me why the authors have disregarded the well-established criteria for identifiying dopamine neurons.

Unfortunately, this intentional or unintentional mistake invalidates the conclusion of these experiments. The firing rate of interneurons within the midbrain may very well change as the DREADD receptor is activated specifically on TH-expressing dopamine neurons. This illustrate the complex synaptic circuit in the midbrain but not that C21 has unspecific effects.

Reviewer #2: The aim of this study was to assess the potency and the selectivity of the Compound 21 (C21) to activate the receptor hM4Di in nigral dopaminergic neurons of TH-Cre rats in vivo. For this, the authors investigate two different concentrations of Compound 21 (1 and 0.5mg/kg) in rats expressing m-cherry (control group) or the receptor hM4Di in both males and females. C21 induced increase in firing rate of nigral neurons at higher concentration in the control group of males that disappears when used at low concentration. In rats expressing the hM4Di receptor, C21 at low concentration induced the expected inhibitory effects. In contrast, C21 at low concentration still induce off-target effects in female control group. Main conclusion of the study is that control groups and animal gender should be always considered when planning chemogenetic studies.

I find the study of potential interest for scientist using chemogenetics tools for investigating behaviors and the underlying neural circuits with relevant information about off-targets effects that clearly can affect data interpretation, strength of the paper. The introduction presents the context and logically raises the main questions of the study and the results and methods sections are generally well organized, following a logic flow. Despite that, this study suffers a number of major and minor weaknesses that need to be addressed.

Major concern:

A) Small sample size. One major limitation in all experiments of this work is the small number of recordings from which the data are generated and conclusions are drawn. Considering the number of variables to compare (two different concentration of C21, control versus transgenic and males versus females), the firing values of the control groups and the variability of the firing between individual of the same group (e.g. different between males and females with the last group having a much larger variability) and the effect size, the number of recordings (the group that has the most is 8) are insufficient to classifying non-significant results as different. This is the case for the experiments on the female groups (both in the control group and the hM4Di-espressing group), one major result and conclusion of the paper. How the authors justify these sample size? Did they run any proper statistical power analysis on this study? Statistical power analysis should be provided to justify the sample sizes. If small sample sizes are justified by power analysis, then proper statistical tests should then be employed (see next point). If power analysis yielded larger sample sizes, then additional experiments should be conducted to achieve an appropriate power.

B) Statistics. Even though the authors indicated in the method section the type of statistical tests used in the paper, this reviewer was unable to locate which data set has been analyze with which statistical test. Further, multiple different types of tests were employed with no justification provided for why using these tests (parametric distribution, similar variability, ect). Third, no statistical power is reported for the statistical significances reported here. Statistical power should be reported whenever statistical significance is reported for every measure. Fourth, with the very small sample sizes used here, t-tests should not be used. Instead, exacts tests should have been used with statistical power reported when statistical significance is achieved. Exacts tests are the proper tests for small sample sizes.

C) The low resolution and the small size of the fonts of the two main figures make for this reviewer impossible to properly read graphs and numbers. The authors must submit figures with appropriate resolution and font size to facilitate data visualization.

Addressing these critical points will give solid base to the results of this study to draw very much needed conclusions.

Minor concerns:

-the general tone of the paper on the results presented here should be tone down. Sentence as ‘We demonstrated here for the first time, to our knowledge’ (e.g. line 163, line 183) should be simplified to ‘here, we demonstrated…or the results indicate…

-Lines 17-18. The introduction of the abstract should be smoother since tens of studies using CNO have been published and the off-targets effects are well defined. For example, the abstract may begin this way: Designer Receptors Exclusively Activated by Designer Drugs (DREADDs) represent a technical revolution in integrative neuroscience. However, the first used ligands exhibited dose-dependent selectivity for their molecular target, leading to unspecific effects.

-Lines 52-54. It is important to mention studies that have identified proper concentration of CNO to reduce off-targets effects.

-lines 80-81: the sentence ‘Before assessing the potential effects of 81 C21 on SNc neuronal activity by using extracellular electrophysiology (Figs 2 and S1)’ should be eliminated since these results will be described in the following sections.

-Line 88 and line 140: the use of the sentence ‘no effect of time whatever the transgene condition’ should be re

6. PLOS authors have the option to publish the peer review history of their article (what does this mean?). If published, this will include your full peer review and any attached files.

Reviewer #1: No

Reviewer #2: No

---

## [Author Response · Author response to Decision Letter 0]

27 Jul 2020

We thank the Reviewers for their encouraging, thoughtful and helpful comments and apologize for some misunderstanding reveal in the light of their comments. Below, a point by point response to the Reviewers’ comments, including additional data and analyses. 

Reviewer 1: 

1. My major concern about this study is whether the authors in fact record from dopamine neurons or not. There is a huge literature on the firing rate of dopamine neurons and the basal firing rate reported is unambiguously between 2-6Hz, depending on the level of anesthesia. In contrast, Goutaudier et al recorded from cells that fire at >20Hz and most likely represent GABA producing interneurons in the midbrain. It is unclear to me why the authors have disregarded the well-established criteria for identifying dopamine neurons.

 We apologize for the misunderstanding about the nature of the recorded neurons in this experiment. Indeed, we did not perform single unit recordings of dopaminergic neurons. The aim of this experiment was to evaluate the long-lasting effect of our injections (over 5 hours) which is technically more challenging when performing those unit recordings. This is why we performed multi-unit recordings, using tungsten electrodes, allowing recording of a neuronal population. The firing rate > 20 Hz is therefore due to a firing rate summation from multiple neurons. To avoid confusion, we have now replaced Hz by events/s.

Because the firing rate criteria cannot be used for neuronal identification as it is a summation of action potentials from different neurons, we used the action potential’s shape, which represents a second important criteria to indicate the nature of the neurons recorded. In multi-unit recordings, dopaminergic neuronal extracellular signals are characterized by a triphasic spike, with a duration greater than 2 ms (Coizet et al., 2006; Ungless and Grace, 2012). 

 We thus performed additional spike analysis indicating we are likely to record from a majority of putative dopamine neurons: 

- The waveform average obtained from each of our recordings are characterized by a triphasic spike, with a duration greater than 2 ms (mean duration = 2.7 +/- 0.06 ms) (in the rebuttal Fig. 1). In multi-unit recordings, we previously showed that this waveform is highly reminiscent of dopaminergic neurons in opposition to GABA neurons, characterized by a biphasic spike with a duration less than 1.5 ms (Coizet et al., 2003).

- As illustrated below (in the rebuttal Fig. 1), there is a low variability in the shape of the waveform average obtained from our recordings (the red lines indicate the standard error, that appear very closed to the mean and difficult to distinguish from it). This indicates this average is from a highly homogeneous population of putative dopaminergic neurons with a long duration triphasic action potentials in majority, and is unlikely including GABA neurons with biphasic short-lasting spikes. This is coherent with the description of SNc, as an area containing a majority of dopamine neuron and a minority of GABA neurons (Nair-Roberts et al., 2008)

- Finally, the large majority of the waveform average fulfills the last criteria to identify dopaminergic neurons using electrophysiology suggested by Ungless and Grace: a duration of > 1.1 ms measured from spike initiation to the maximal negative phase of the action potential. This duration was 1.2 +/- 0.03 ms (mean ± SEM) in our experiment (Ungless and Grace, 2012).

Figure 1: Representative example of the waveform average realized on our recording. The total duration of the triphasic spike is indicated in blue. The duration from the spike initiation to the maximal negative phase of action potential is indicated in green. Data are presented as the waveform average +/- SEM

We had clarified this point, in different parts of the “Materials and methods” and “Figure Legend” sections of the manuscript (indicated in red). In addition, we replaced “Firing rate” by “Multi-unit activity” in Fig. 2 of the manuscript and elsewhere in the text.

Unfortunately, this intentional or unintentional mistake invalidates the conclusion of these DREADD receptor is activated specifically on TH-expressing dopamine neurons. This illustrate the complex synaptic circuit in the midbrain but not that C21 has unspecific effects.

 The first part of the Reviewer’s comment pointed out whether or not we were recording DA neurons in this study. We hope this point is now clarified. The second part pointed out whether or not we observed unspecific effects with C21. In the manuscript, we compared, in male ♂-mCherry rats, two doses of C21 (0.5 mg/kg and 1 mg/kg). At 1 mg/kg, C21 induced a robust and long-lasting increase of SNc neurons compared to vehicle and the 0.5 mg/kg condition. Considering that C21, at 1 mg/kg, strongly modified the SNc neurons activity while animals do not express the DREADD, and that this effect disappear at 0.5 mg/kg or is absent in animals treated only with NaCl, it was therefore difficult to conclude anything other than C21 has, at 1 mg/kg in Long-Evans rats, unspecific effects.

 To conclude on this unspecific effect, we also want to take this opportunity to remind that we never conclude on the type of neurons responsible for this effect. Rather, we proposed two main hypotheses by which the potential target of C21 could be directly localized on DA neurons or an alternative and indirect mechanism by which the target of C21 would be localized on GABA neurons. It is critical for us to not over-interpret the results but to warn scientist using C21 about possible side-effects of this compound, whatever are the underlying mechanisms.

Reviewer 2: 

A) Small sample size. One major limitation in all experiments of this work is the small number of recordings from which the data are generated and conclusions are drawn. Considering the number of variables to compare (two different concentration of C21, control versus transgenic and males versus females), the firing values of the control groups and the variability of the firing between individual of the same group (e.g. different between males and females with the last group having a much larger variability) and the effect size, the number of recordings (the group that has the most is 8) are insufficient to classifying non-significant results as different. This is the case for the experiments on the female groups (both in the control group and the hM4Di-espressing group), one major result and conclusion of the paper. How the authors justify these sample size? Did they run any proper statistical power analysis on this study? Statistical power analysis should be provided to justify the sample sizes. If small sample sizes are justified by power analysis, then proper statistical tests should then be employed (see next point). If power analysis yielded larger sample sizes, then additional experiments should be conducted to achieve an appropriate power.

 We apologize for a potential lack of clarity, but 8 is the maximum of animals use per group and not the maximum number of recordings. As 2 recordings were possible per animal (one per side), the number of recordings was between 7 and 10 which is in the range or even above the sample sizes usually published with this kind of techniques and statistical analysis (n = 4, Takasu et al., 2013; n = 8, Gremel and Costa, 2013; n = 6, Fifel et al., 2018). We have tried to indicate and explain in different sections of our manuscript (In the manuscript: Fig 1. – page 7, lines 146-147; Fig 2. – page 8, lines 164-168; Materials and Methods – page 11, line 203; page 12, lines 234-237; page 13, lines 249-250; pages 15, lines 297-301), the number of animals and recording used for each group, to be as transparent as possible.

 We understand the point of the Reviewer. However, we did not run any a priori statistical power analysis as you need to inject in the analysis, the difference you expect to observe between our groups and we had no prediction about the effect we will obtain, especially for the off-target effect. In addition, estimation or computation of power remains controversial (for instance, Prism, the statistical software used for this study and frequently used in biology, decided to not calculate power for these reasons), especially as it is directed related to the p value (Goodman and Berlin, 1994; Hoenig and Heisey, 2001). This is why we have decided before the experiment, that we will calculate and provide partial η2 values when it will be possible, as an alternative and reliable indicator of the robustness or not, of our effects (Levine and Hullett, 2002), as we did recently (Magnard et al., 2018). This approach allowed us to found a strong effect size for the treatment x time interaction for male (in the manuscript: 0.44, page 6, line 117) but not for female (in the manuscript: 0.19, page 7, line 132).

 To answer the question of the Reviewer, we ran with another software (SigmaStat) a power analysis on our results to give as estimation of the requested sample sizes. For the off-target effect in male (in the manuscript: Fig. 2D, middle panel), for an alpha set at 0.05, the power of the ANOVA was estimated to be 0.98, giving a sample size of 4 to detect the effect as significant with a power of 0.80, and confirms the robustness of this effect. For the DREADDs-mediated effect in male (in the manuscript: Fig. 2E, middle panel), the power was estimated to be 0.94, giving a sample size of 5. For the DREADDs-mediated effect in female (in the manuscript: Fig. 2F, middle panel), the power that was estimated was really low (0.11), and a minimum sample size of 15 will be necessary for a power of 0.80. See the results of the analyses for the sample sizes just below:

Off target effect, male (Fig. 2D):

Sample Size for ANOVA: jeudi, juin 25, 2020, 14:43:35

Data source: Data 1 in Notebook 1

Sample Size 4,000

Difference in Means 119,000

Standard Deviation 40,000

Number of Groups 4

Power 0,800

Alpha 0,0500

DREADDs-mediated effect, male (Fig. 2E):

Sample Size for ANOVA: jeudi, juin 25, 2020, 14:39:13

Data source: Data 1 in Notebook 1

Sample Size 5,000

Difference in Means 33,000

Standard Deviation 14,000

Number of Groups 4

Power 0,800

Alpha 0,0500

DREADDs-mediated effect, female (Fig. 2F):

Sample Size for ANOVA: jeudi, juin 25, 2020, 14:46:48

Data source: Data 1 in SampleSize-PLOSONE

Sample Size 15,000

Difference in Means 27,000

Standard Deviation 21,000

Number of Groups 4

Power 0,800

Alpha 0,0500

 The objective and the message for the female was not to say that the effect is different than in male but, as written (in the manuscript: page 7, lines 133-136), and as it can be observed (in the manuscript: Fig. 2F, left panel), that a residual off-target effect may mitigated the DREADDs-mediated effect that we recognized to be also present in female. In order to clarify our message, we propose to add at the end of the results section: “Therefore, detecting a statistically significant DREADDs-mediated effect may require a greater number of animals for female with this dose of C21”.

 For the last point, the use of parametric analyses appears justified as they are standardly used in the field with this range of sample sizes (e.g., Coizet et al., 2006; Fifel et al., 2018; Gremel and Costa, 2013). In addition, we have verified the assumptions of normality (Shapiro-Wilk and Kolmogorov-Smirnov tests) and sphericity (Bartlett's test). We apologize for not reporting this point in the first version of the manuscript. This is now specified in the Data analyses section.

B) Statistics. Even though the authors indicated in the method section the type of statistical tests used in the paper, this reviewer was unable to locate which data set has been analyze with which statistical test. Further, multiple different types of tests were employed with no justification provided for why using these tests (parametric distribution, similar variability, ect). Third, no statistical power is reported for the statistical significances reported here. Statistical power should be reported whenever statistical significance is reported for every measure. Fourth, with the very small sample sizes used here, t-tests should not be used. Instead, exacts tests should have been used with statistical power reported when statistical significance is achieved. Exacts tests are the proper tests for small sample sizes.

 We have now reported the type of test for each result in the corresponding section and completed the justification of their use in the Data analyses section (see also point A).

For the questions concerning statistical power and sample sizes, see also point A. As the distribution appear parametric, it seems justify to use t-tests to be in line with the ANOVA analyses. Again, t-tests are also standardly used in the field in publication with similar sample sizes (Coizet et al., 2006, 2003; Fifel et al., 2018; Gremel and Costa, 2013; Paul et al., 2017; Takasu et al., 2013). To confirm the statistically significance of our effects and ensure that we did not have incidentally increase our alpha risk above 5%, we however ran non-parametric tests. A Wilcoxon matched-pairs signed rank test to compare the period in which the dose of 1 mg/kg of C21 produced the increase in neuronal activity to the vehicle period (in the manuscript: Fig. 2D, right panel) found a significant and exact p value of 0.0313. A similar test to compare the period in which the dose of 0.5 mg/kg of C21 produced the decrease in neuronal activity to the vehicle period (in the manuscript: Fig. 2E, right panel) found a significant and exact p value of 0.0098. The results of the other test were non-significant, providing similar results as the ones obtained with parametric t-tests.

 Taking together, the complementary analyses of point A and B consistently support our prior analyses, indicating that the off-target effect of 1 mg/kg of C21 is robust, and that it is easier to statistically detect the DREADDs-mediated effect of 0.5 mg/kg of C21 in male than in female. 

C) The low resolution and the small size of the fonts of the two main figures make for this reviewer impossible to properly read graphs and numbers. The authors must submit figures with appropriate resolution and font size to facilitate data visualization.

 We apologize for this technical issue that we hope we have fixed for the submission of the revised version of our manuscript.

Minor concerns:

-the general tone of the paper on the results presented here should be tone down. Sentence as ‘We demonstrated here for the first time, to our knowledge’ (e.g. line 163, line 183) should be simplified to ‘here, we demonstrated…or the results indicate…

-Lines 17-18. The introduction of the abstract should be smoother since tens of studies using CNO have been published and the off-targets effects are well defined. For example, the abstract may begin this way: Designer Receptors Exclusively Activated by Designer Drugs (DREADDs) represent a technical revolution in integrative neuroscience. However, the first used ligands exhibited dose-dependent selectivity for their molecular target, leading to unspecific effects.

-Lines 52-54. It is important to mention studies that have identified proper concentration of CNO to reduce off-targets effects.

-lines 80-81: the sentence ‘Before assessing the potential effects of 81 C21 on SNc neuronal activity by using extracellular electrophysiology (Figs 2 and S1)’ should be eliminated since these results will be described in the following sections.

-Line 88 and line 140: the use of the sentence ‘no effect of time whatever the transgene condition’ should be re

 We thank the Reviewer for these suggestions. Modifications have been throughout the manuscript accordingly.

 

Reference: 

Björklund A, Dunnett SB (2007) Dopamine neuron systems in the brain: an update. Trends Neurosci 30:194–202.

Coizet V, Comoli E, Westby GWM, Redgrave P (2003) Phasic activation of substantia nigra and the ventral tegmental area by chemical stimulation of the superior colliculus: An electrophysiological investigation in the rat. Eur J Neurosci 17:28–40.

Coizet V, Dommett EJ, Redgrave P, Overton PG (2006) Nociceptive responses of midbrain dopaminergic neurones are modulated by the superior colliculus in the rat. Neuroscience 139:1479–1493.

Fifel K, Meijer JH, Deboer T (2018) Circadian and Homeostatic Modulation of Multi-Unit Activity in Midbrain Dopaminergic Structures. Sci Rep 8:1–14.

Goodman SN, Berlin JA (1994) The use of predicted confidence intervals when planning experiments and the misuse of power when interpreting results. Ann Intern Med.

Gremel CM, Costa RM (2013) Orbitofrontal and striatal circuits dynamically encode the shift between goal-directed and habitual actions. Nat Commun.

Hoenig JM, Heisey DM (2001) The abuse of power: The pervasive fallacy of power calculations for data analysis. Am Stat.

Levine TR, Hullett CR (2002) Eta Squared, Partial Eta Squared, and Misreporting of Effect Size in Communication Research. Hum Commun Res.

Magnard R, Vachez Y, Carcenac C, Boulet S, Houeto JL, Savasta M, Belin D, Carnicella S (2018) Nigrostriatal dopaminergic denervation does not promote impulsive choice in the rat: Implication for impulse control disorders in Parkinson’s disease. Front Behav Neurosci 12:1–10.

Nair-Roberts RG, Chatelain-Badie SD, Benson E, White-Cooper H, Bolam JP, Ungless MA (2008) Stereological estimates of dopaminergic, GABAergic and glutamatergic neurons in the ventral tegmental area, substantia nigra and retrorubral field in the rat. Neuroscience 152:1024–1031.

Paul R, Choudhury A, Kumar S, Giri A, Sandhir R, Borah A (2017) Cholesterol contributes to dopamineneuronal loss in MPTP mouse model of Parkinson’s disease: Involvement of mitochondrial dysfunctions and oxidative stress. PLoS One 12:1–22.

Takasu NN, Pendergast JS, Olivas CS, Yamazaki S, Nakamura W (2013) In Vivo Monitoring of Multi-Unit Neural Activity in the Suprachiasmatic Nucleus Reveals Robust Circadian Rhythms in Period1-/- Mice. PLoS One 8:1–8.

Ungless MA, Grace AA (2012) Are you or aren’t you? Challenges associated with physiologically identifying dopamine neurons. Trends Neurosci.

---

## [Decision Letter · Decision Letter 1]

11 Aug 2020

Compound 21, a two-edged sword with both DREADD-selective and off-target outcomes in rats

PONE-D-20-14375R1

Dear Dr. Carnicella,

We’re pleased to inform you that your manuscript has been judged scientifically suitable for publication and will be formally accepted for publication once it meets all outstanding technical requirements.   

Kind regards,

Gilberto Fisone

Academic Editor

PLOS ONE

Additional Editor Comments (optional):

I encourage the authors to implement the last recommendations expressed by the referees (see Review Comments to the Author)

Reviewers' comments:

Reviewer's Responses to Questions

**Comments to the Author**

Reviewer #1: All comments have been addressed

Reviewer #2: All comments have been addressed

2. Is the manuscript technically sound, and do the data support the conclusions?

Reviewer #1: Yes

Reviewer #2: Yes

3. Has the statistical analysis been performed appropriately and rigorously? 

Reviewer #1: Yes

Reviewer #2: Yes

4. Have the authors made all data underlying the findings in their manuscript fully available?

Reviewer #1: Yes

Reviewer #2: Yes

5. Is the manuscript presented in an intelligible fashion and written in standard English?

Reviewer #1: Yes

Reviewer #2: Yes

6. Review Comments to the Author

Reviewer #1: Thank you for clarifying how the identification of the neurons are performed. I recommend that this information is added to the Figure 2. Although some readers are aware of these criteria, I believe it will guide a broader audience and avoid confusion if supplemental fig 3 is shown in the main part of the manuscript. My concern has been addressed in this version of the manuscript and I have no further comments.

Reviewer #2: The authors answered posivitely all reviewers comments. It may be usefull if the authors create a supplementary table with all statistical tests, significance and power analysis . I believe it is important to make this infomration accessible to the reader since the type of data and their interpretation crucially depend by the comparisosn of the different groups.

7. PLOS authors have the option to publish the peer review history of their article (what does this mean?). If published, this will include your full peer review and any attached files.

Reviewer #1: No

Reviewer #2: No

---

## [Editor Report · Acceptance letter]

9 Sep 2020

PONE-D-20-14375R1 

Compound 21, a two-edged sword with both DREADD-selective and off-target outcomes in rats. 

Dear Dr. Carnicella:

I'm pleased to inform you that your manuscript has been deemed suitable for publication in PLOS ONE. Congratulations! Your manuscript is now with our production department. 

Kind regards, 

on behalf of

Dr. Gilberto Fisone 

Academic Editor

PLOS ONE